# An Experimental Study on Color Shift of Injection-Molded Mobile LGP Depending on Surface Micropattern

**DOI:** 10.3390/polym12112610

**Published:** 2020-11-06

**Authors:** Mooyeon Kim, Junhan Lee, Kyunghwan Yoon

**Affiliations:** Department of Mechanical Engineering, Dankook University, Yongin 16890, Korea; mooyeon@dankook.ac.kr (M.K.); junhan0526@dankook.ac.kr (J.L.)

**Keywords:** injection molding, LGP, CIE xy, direct transmittance, total transmittance, color shift, surface micropattern, scattering, yellow shift

## Abstract

In the display industry, the LCD backlight unit (BLU) module is variously used in mobile phones, notebook computers, monitors, and TVs. The light guide plate (LGP), which is one of the core parts of a BLU, is getting bigger and thinner consistently. Conventional injection methods and injection processes like injection compression molding (ICM) are becoming more complex and harsher with high-speed injection at high mold and melt temperatures. These approaches lead to a change in physical properties and a decrease in optical properties such as yellowing and color shift in injection-molded parts. In the present study, an injection molding experiment was conducted to understand the effect of surface patterns and major injection process conditions like mold and melt temperatures on the color shift in injection-molded mobile LGP. Optical properties obtained by the direct and total transmittance and CIE xy chromaticity diagram for injection-molded mobile LGP were measured with a UV–visible spectrophotometer. From the measurement of patternless LGP, it was found that total or direct transmittance was not affected by molding process variables. It was also found that yellow shift, ΔE(xy), occurred as much as 0.00111 ± 0.00014, and a color shift angle, Θ(xy), of 43.05 ± 2.44° was recorded in CIE coordinates for all nine experimental cases. From the measurement of total transmittance of patterned LGP, ΔE(xy) and Θ(xx) were found to be almost the same as those of patternless LGP for the locations of low and medium density of the pattern for the LGP, T1 and T2. The measured data of direct transmittance of patterned LGP showed that additional yellow shift due to scattering caused by surface micropattern. Interestingly, Θ(xy) of patterned data remained 43.05 ± 2.44°, which was almost the same as that found in the case of patternless LGP.

## 1. Introduction

Great progress has been made in the display market, as seen by the technological evolution from the cathode ray tube (CRT) to the recently developed organic light-emitting diode (OLED). Since OLED displays can be used in large TVs and smartphones, their use is gradually increasing. However, there are still difficulties faced in terms of price competitiveness. For this reason, LCD panels requiring backlight units (BLUs) are still the most popular panel technology. Figure 1 shows the structure of a BLU. A BLU consists of a light source, optical films, and a light guide plate (LGP), and LGP includes micropatterns to control the optical path. The LGP plays the most important role in converting a point light source into a surface light source, as shown in Figure 1.

Ju et al. studied the optical characteristics of the viewing angle through optical analysis in relation to the use of the prism film, which is an optical film applied to the BLU module [1], and Joo et al. studied the change in luminance and uniformity through optical analysis considering the transferability of the LGP [2]. In addition, many studies have been actively conducted in terms of molding, mold, and material with the aim of reducing LGP thickness due to the trends toward larger screen size and thinner displays. Yokoi et al. made a vacuum in the mold and observed the filling process while conducting a molding experiment to increase the transferability of the V-groove pattern in the ultrahigh-speed injection molding process [3]. In addition, Hong et al. conducted a comparative study on the transferability and birefringence distribution of the pattern through 7-inch LGP molding using the rapid injection–compression molding (RICM) process that combines the injection–compression process and rapid mold heating [4]. However, previous studies related to LGP mainly focused on optimization of optical patterns to increase luminance and uniformity and enhance the transcriptional properties of micropatterns during molding [5,6,7,8,9,10,11,12]. Interest in color reproducibility has recently started to increase, as this factor allows implementing more natural images while competing with display technologies such as OLED. Min et al. introduced the concept of color shift angle and studied the effect of molding processes related to melt temperature, injection speed, mold temperature, injection speed, and packing pressure on the color difference of an LGP and the direction and magnitude of the color shift [13]. However, they could not determine the contribution to the color shift made by the micropatterns, which were manufactured for uniform distribution of the light from the light source in the LGP to the entire LCD panel. In the present study, the effect of surface patterns and the injection molding process on the change in transmittance and color shift of the mobile LGP was investigated.

## 2. Theory

### 2.1. Color Chromaticity

It is very difficult to accurately detect an objectified color as a human sense. Historically, there have been many studies quantifying colors such as red, blue, and yellow. In the past in art history, color was quantified using the Munsell color system developed by Munsell; however, in the digital age, there are limitations in applying this system to color processing [14]. In the late 1920s, Wright [15] and Guild [16,17] defined a standard observer and a color matching function through a color matching experiment. In 1931, the international commission on illumination (CIE) announced the standard color space, which is a CIEXYZ colorimetric space. CIE defined colors as numbers by designating three elements as standard observer, lighting, and color matching functions as follows:(1)$$X=\int_{380}^{780}\bar{x}\left(\lambda \right)S\left(\lambda \right)d\lambda$$
(2)$$Y=\int_{380}^{780}\bar{y}\left(\lambda \right)S\left(\lambda \right)d\lambda$$
(3)$$Z=\int_{380}^{780}\bar{z}\left(\lambda \right)S\left(\lambda \right)d\lambda$$
where *S*(*λ*) is an intensity value according to a wavelength obtained from a light spectral distribution, and $\bar{x}$(*λ*), $\bar{y}$(*λ*), and $\bar{z}$(*λ*) are the color matching functions. In this way, the *x* and *y* values of color coordinates are obtained through normalization from the tristimulus values by Equations (4)–(6).
(4)$$x=\frac{X}{X+Y+Z}$$
(5)$$y=\frac{Y}{X+Y+Z}$$
(6)$$z=\frac{Z}{X+Y+Z}=1-x-y$$

The *x* and *y* values obtained in this way can be expressed in the CIE xy diagram and are called the CIE1931 color space. The color difference between two points in CIE1931, ΔE(xy), is defined in Equation (7) and shown in Figure 2.
(7)$$\Delta E\left(xy\right)=\sqrt{{\left(x_{1}-x_{0}\right)}^{2}+{\left(y_{1}-y_{0}\right)}^{2}}$$

The color shift angle of point 1 from point 0, Θ(xy), is defined as in Figure 2. Since 1931, many color spaces have been released depending on the purpose of use, but CIEXYZ is the origin of all color quantification and can be converted to other color spaces. The spectrophotometer used in this study has the reference point of white light, i.e., (*x*_0_, *y*_0_) = (0.31001, 0.31623), in CIE1931 xy color space.

### 2.2. Two Types of Total Transmittance Measurement

One of the most important parameters in transparent displays is the transmittance. For the plates of transparent material or medium, the transmittance is usually defined as the percentage ratio between the spectral intensities of incident (*I_0_*) and transmitted ray (*I*) as shown in Equation (8).
(8)$$T\;\left(\%\right)=\;\frac{I}{I_{0}}\times 100$$

## 3. Experiments

### 3.1. Material and Molding Equipment

PMMA and PC are mainly used as the material of an LGP. In the present experiment, Mitsubishi Iupilon HL-4000 (PC) was used as this molding material. HL-4000 is widely used in LGP molding due to its high impact strength and heat resistance and especially its high fluidity. Table 1 shows the material properties of Iupilon HL-4000. The LGP mold shown in Figure 3a is a two-stage mold and has two cavities of 79.08 mm in length, 51.72 mm in width, and 0.43 mm in nominal thickness, as shown in Figure 3b. Patterned LGP has a denser pattern from the flow end to the gate, which enables a uniform light distribution to be obtained. In this case, micropatterns of 40 µm in diameter and 4.5 µm in height were manufactured on the fixed side of the mold by the diamond processing method. In the case of patternless LGP, the same material and the same mirrored surface as used in the patterned LGP were manufactured without the micropattern. In the molding experiment, FANUC’s injection molding machine (ROBOSHOT α-S250iA, FANUC, Bucheon, Korea) was used, as shown in Figure 4. The general specifications of the injection molding machine used in the experiment are summarized in Table 2.

### 3.2. Experimental Conditions

In order to reduce the change of flow index due to moisture in the material during molding, the material was processed under the drying conditions recommended by the manufacturer. In order to obtain high fluidity, the melting temperature and injection speed used in the range of the molding window were rather high in comparison to those used in conventional injection molding conditions. In order to check the change in optical properties according to the molding conditions, the melting temperature, mold temperature, injection speed, and packing pressure were set as processing parameters, and these factors were arrayed by L9 for statistical analysis in the design of experiments (DOE) test set. Detailed processing conditions and levels applied to produce patterned and the patternless LGPs are shown in Table 3 and Table 4.

### 3.3. Measurement of Optical Properties

Transmittance generally means total transmittance, which is divided into direct transmittance and diffuse (or scattered) transmittance, as shown in Figure 5a [18]. Direct transmittance excludes the intensity values of the scattered portion from the patterned surface, as shown in Figure 5b.

To evaluate the optical properties, total transmittance and direct transmittance were measured in the visible region (380–780 nm) using Agilent’s Cary 5000 UV-Vis-NIR spectrometer. It can measure the transmittance versus wavelength from 3300 to 175 nm using the halogen (visible range) and deuterium lamps (UV range). General specifications of the spectrometer are listed in Table 5. Direct transmittance was measured for six points (points 1 to 6) from the gate of the 3.5-inch LGP to the flow end, as shown in Figure 6a, and total transmittance was measured for three areas (T1, T2, and T3) along the center line of the 3.5-inch LGP, as shown in Figure 6b. For each case number, 20 samples were taken, and optical properties including transmittance, YI level, and color value based on CIE1931 xy diagram were evaluated.

## 4. Results

The injection process conditions shown in Table 4 were applied to mold the patternless and patterned LGP samples. The distribution of color coordinate values in the injection-molded patternless and patterned LGP samples was experimentally obtained, and the effect of the injection molding process and surface micropattern on the color shift was investigated.

### 4.1. Measurement of Color Coordinates from Total Transmittance

Firstly, the color coordinates for CIE1931 color space of LGP were obtained from the total transmittance measurement set-up, as shown in Figure 3a. Figure 7a,b show the distribution of color coordinates for patternless and patterned samples, respectively, including the data of all the locations (T1, T2, and T3) for Case Nos. 1, 4, and 7. All points of the patternless LGP shown in Figure 7a show the yellow shift value, ΔE(xy)*,* of 0.00111 ± 0.00014 and the color shift angle, Θ(xy), of 43.05 ± 2.44°.

The color coordinates of T1 and T2 for the patterned LGP shown in Figure 7b show nearly the same range of ΔE(xy), i.e., 0.00111 ± 0.00014, as patternless data shown in Figure 3a. However, in the case of T3, the measurement error increases due to the very high density of the surface micropattern. The measured transmittance value in visible wavelength was rather small, and ΔE(xy) and Θ(xy) for T3 showed deviation from the data of T1 or T2, as shown in Figure 7b.

### 4.2. Measurement of Color Coordinates from Direct Transmittance

Figure 8a shows the color coordinates of the patternless LGP obtained by measuring direct transmittance for all nine cases in Table 4. The yellow shift value, *ΔE(xy)*, of 0.00106 ± 0.00011 and color shift angle, Θ(xy), of 39.94 ± 3.03° were obtained from data regression. *ΔE(xy)* in Figure 8a shows almost the same color shift value as in Figure 7a for patternless LGP and T1 and T2 data in Figure 7b for patterned LGP.

Figure 8b shows the distribution of color coordinates for all nine cases obtained by direct transmission measurements of patterned LGP. The yellow shift value shown in Figure 8b increased as the measured point changed from the flow end to the gate, and the fitted color shift angle Θ(xy) was 43.98 ± 1.14°. Further explanation is provided in Section 4.3 with a data set of Case No. 5 as an example.

To ascertain the direction of color shift, i.e., whether it is yellow or blue, the yellow color area is displayed in the CIE1931 color space in Figure 9. If the color coordinate shifts with a color shift angle between 22.18° (to the 590 nm direction) and 60.78° (to the 560 nm direction) from the light source, it means that the color changes to yellow after passing the transparent sample. In patterned LGP, because the color shift angle measured from direct transmittance was 43.98 ± 1.14°, it can be concluded that a definite linear yellow shift phenomenon occurs after passing LGP samples.

### 4.3. Color Coordinate Data of Case No. 5

Figure 10 shows the color coordinates of the patternless and patterned LGPs obtained from measuring direct transmittance for Case No. 5 in Table 4. Case No. 5 is shown as an example of detailed data extracted from Figure 8a,b. In the case of the patternless LGP, color shift values measured by direct transmittance were less than those of patterned data. The yellow shift value, *ΔE(xy)*, of 0.00106 ± 0.00011 can be obtained, as shown in Figure 7a. It can be concluded that the effect of color difference is little regardless of different molding conditions. This yellow shift value of 0.00106 ± 0.00011 is recorded as *ΔE_0_* in Figure 10. *ΔE_0_* can be explained as the amount of yellow shift occurring after a path of the light through the thickness direction of the LGP sample, which can be attributed to the injection molding process.

The color coordinates of the patterned LGP of Case No. 5 shown in Figure 10 lie in the color shift angle, Θ(xy), of 43.79 ± 1.00° within the error range of Θ(xy) in Figure 7b. From the flow end (Point 1) area to the gate area (Point 6), the color coordinates exhibited a more yellowish shift along a straight line from the reference.

The extra amounts of yellow shift, i.e., *ΔE_1_, ΔE_3_*, and *ΔE_6_* for different locations in Figure 10, could be the results generated by scattering or other phenomena. As a next step, the image of the laser pattern was observed assess these influences.

Figure 11c shows the structure of the microdotted pattern of patterned LGP measured by optical surface inspection with μ-surf (Nanofocus, Inc., Glen Allen, VA, USA; Figure 11a). From the flow end to the gate, microdotted pattern with a diameter of 40 μm and a height of 4.5 μm was manufactured with increasing density on the fixed mold surface. The scattering images for the different positions of the patterned LGP were investigated using the vision equipment of Figure 11b and a He-Ne Laser. Only the bright spot near the center of the transmitted image in Figure 11c is measured in direct transmittance. As the density of the microdotted patten increased, more scattering-caused portions of extra yellow shift, i.e., *ΔE_1_*, *ΔE_3_*, *ΔE_6_*, were generated, as shown in Figure 10. From this investigation, a simple inspection system of surface-micropatterned LGPs used in the industry, such as the inspection with the naked eye shown in Figure 11d, can detect distortions in the amount of yellow shift. When the observer sees the patterned LGP (direct transmittance evaluation shown in Figure 11d), the amount of yellow shift can be observed to be greater than that seen in the case of the pattternless transparent product (as explained in Figure 10).

## 5. Conclusions

In a previous study, Min et al. tried to explain the cause of the color shift in patterned LGP as the difference in residence time during the filling process of injection molding [13]. In the present study, the effect of the existence of the LGP’s micropattern on the evaluation of transmittance and color shift was experimentally investigated. Total transmittance test showed consistent values of the yellow shift, ΔE(xy), of 0.00111 ± 0.00014 regardless of different molding conditions. This one path yellow shift value is restricted to the specific experiment and specific pattern density of T1 and T2, but not T3. The transmittance data obtained from the simple inspection system of patterned LGP, similar to direct transmittance set-up, could mislead observers regarding the amount of yellow shift in injection-molded transparent products. The contribution to the yellowing can be divided into two parts: one from the injection molding process, and the other from scattering or other optical phenomena. Interestingly, the shift angles of both contributions were found to be almost identical in error range.

## Figures and Tables

**Figure 1 polymers-12-02610-f001:**
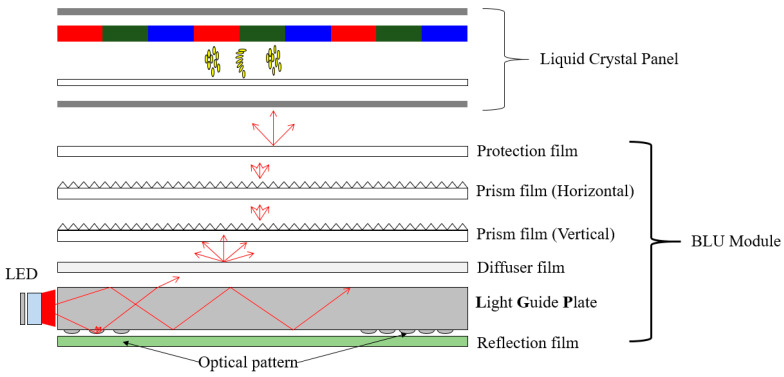
The structure of a backlight unit (BLU).

**Figure 2 polymers-12-02610-f002:**
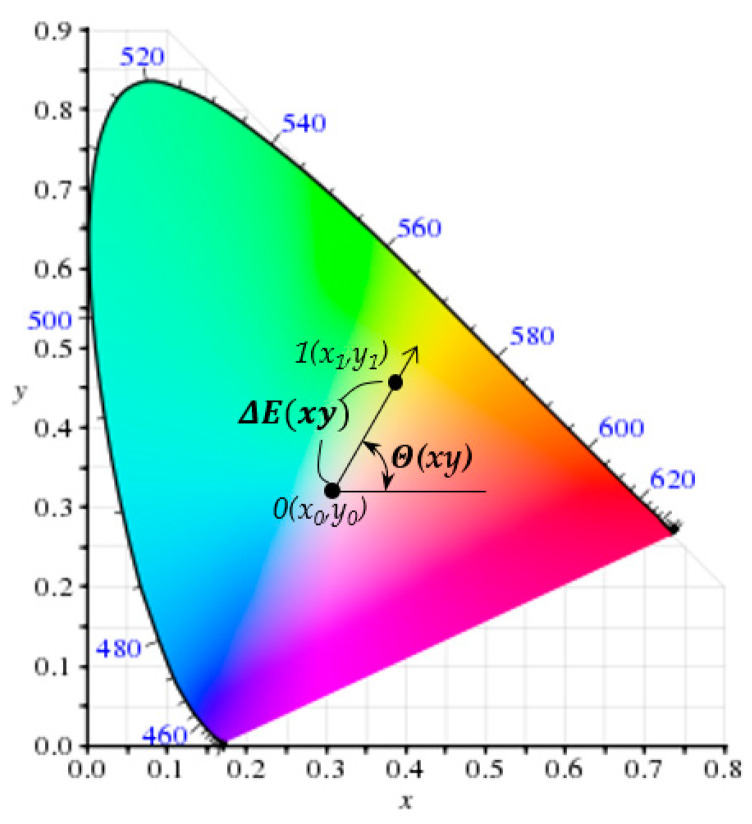
The color space of CIE1931.

**Figure 3 polymers-12-02610-f003:**
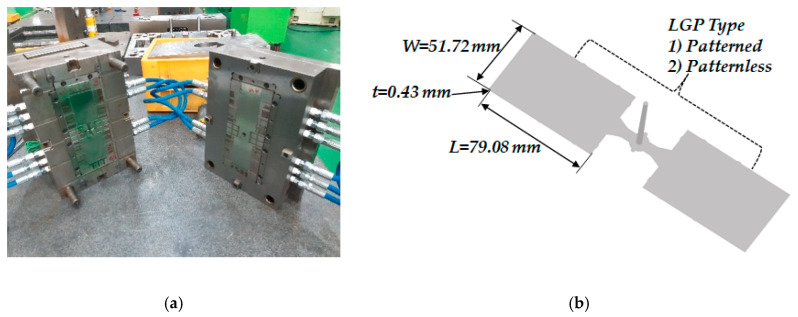
(**a**) Depiction of the mold; (**b**) light guide plate (LGP) dimensions.

**Figure 4 polymers-12-02610-f004:**
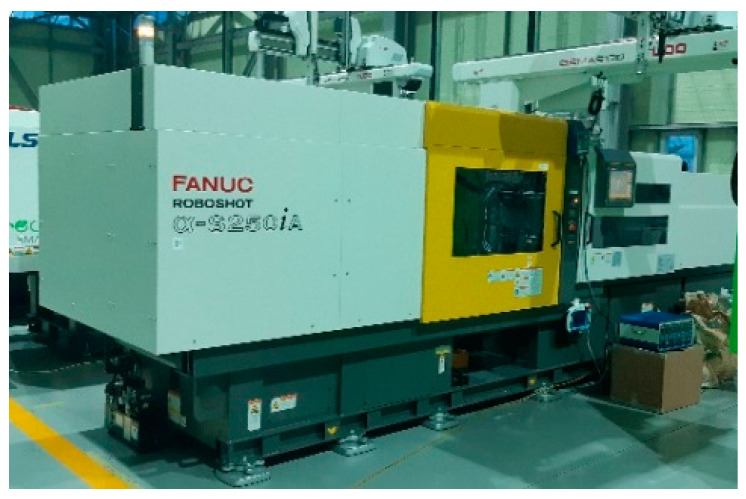
A photograph of injection molding machine used (ROBOSHOT α-S250iA, FANUC).

**Figure 5 polymers-12-02610-f005:**
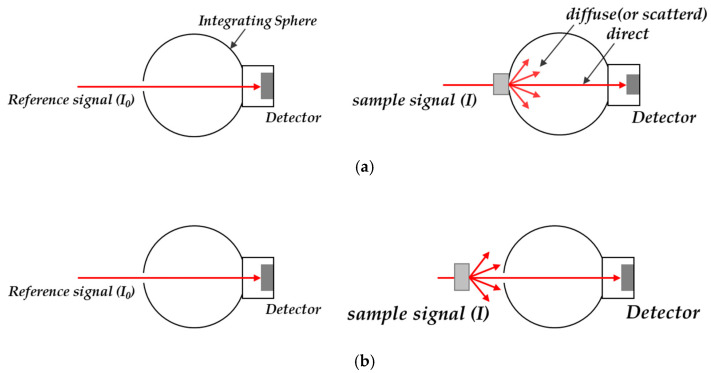
Schematic diagrams for the measurement of (**a**) total transmittance and (**b**) direct transmittance.

**Figure 6 polymers-12-02610-f006:**
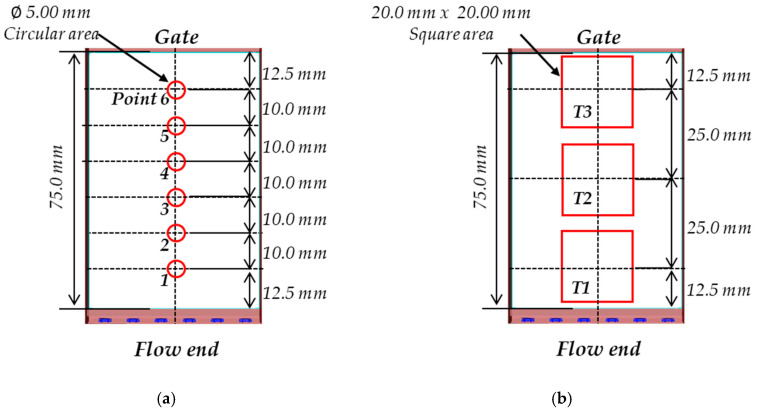
Measurement points for (**a**) direct transmittance and (**b**) total transmittance.

**Figure 7 polymers-12-02610-f007:**
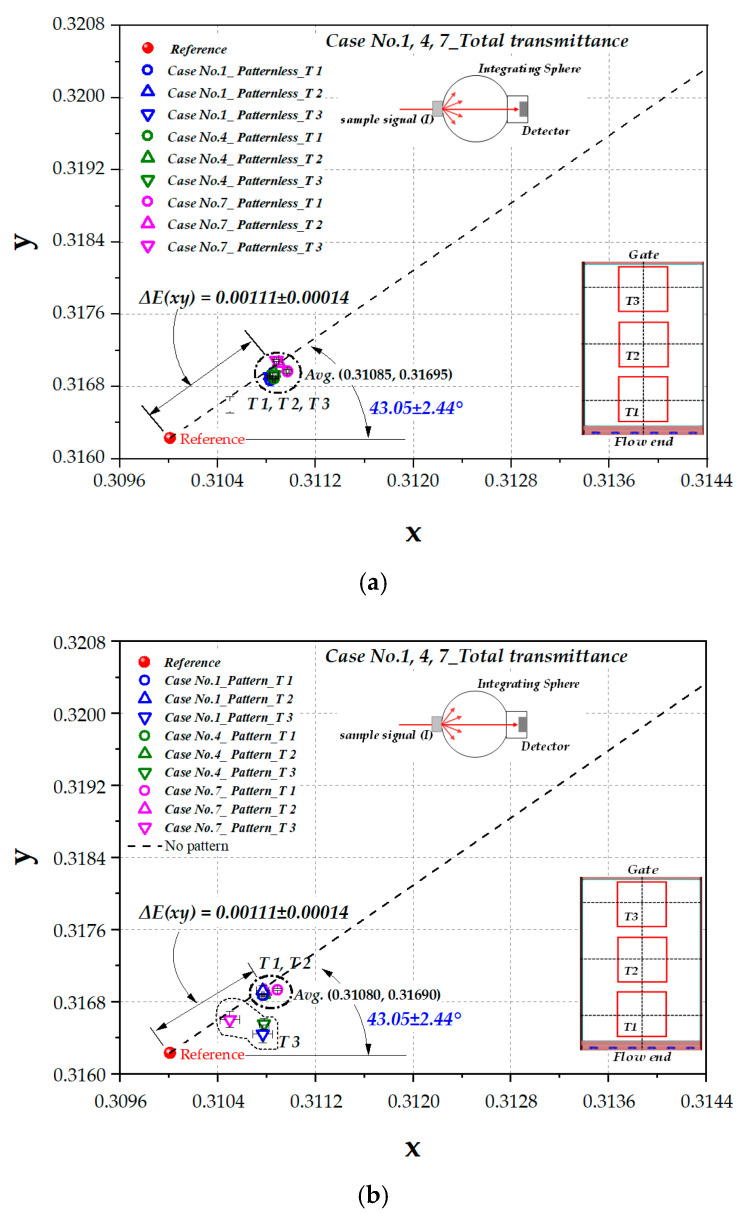
The representation of color shift from total transmittance shown in CIE1931 color space: (**a**) patternless LGP (Case Nos. 1, 4, and 7); (**b**) patterned LGP (Case Nos. 1, 4, and 7).

**Figure 8 polymers-12-02610-f008:**
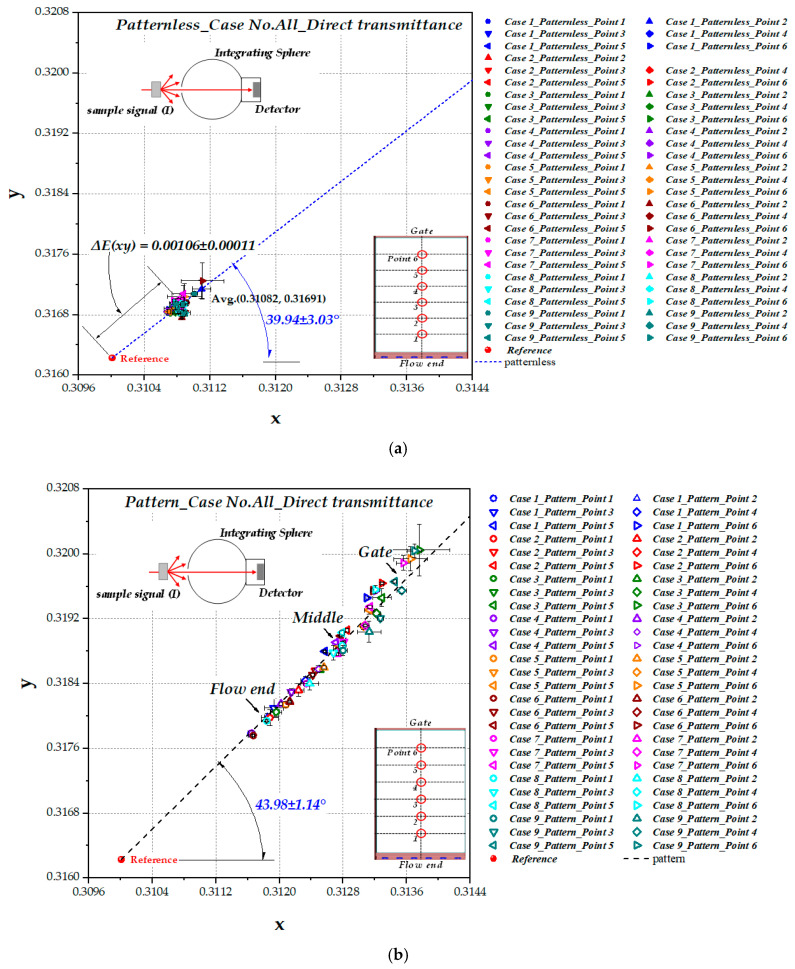
The representation of color shift from direct transmittance shown in CIE1931 color space: (**a**) patternless LGP (all 9 cases); (**b**) patterned LGP (all 9 cases).

**Figure 9 polymers-12-02610-f009:**
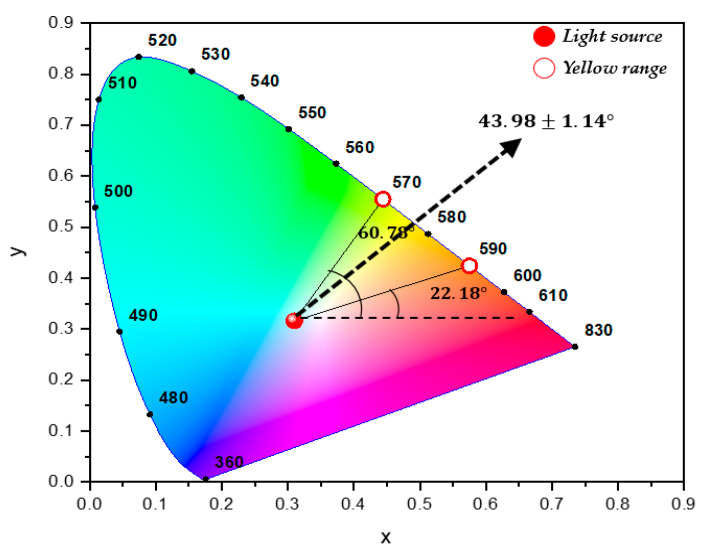
The representation of color shift from direct transmittance at 3 measurement locations (yellow range is shown in CIE1931 xy color diagram).

**Figure 10 polymers-12-02610-f010:**
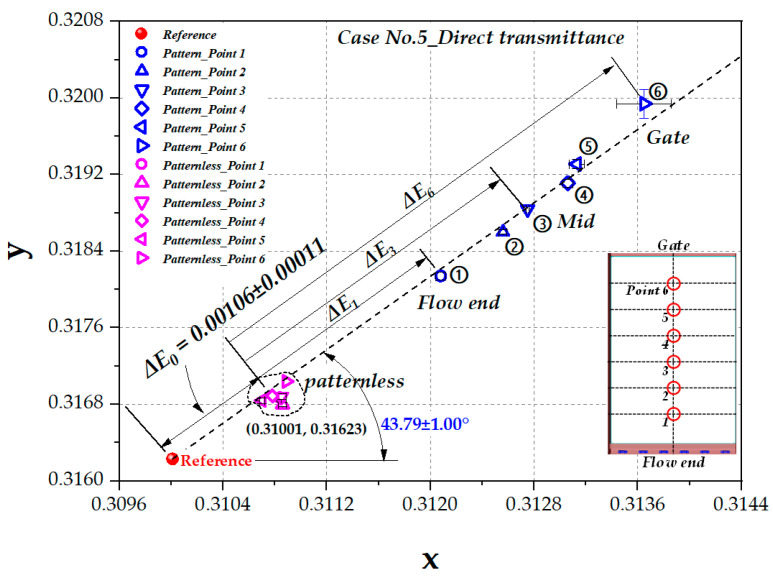
The representation of color shift from direct transmittance shown in CIE1931 color space of patternless and patterned LGPs (Case No. 5).

**Figure 11 polymers-12-02610-f011:**
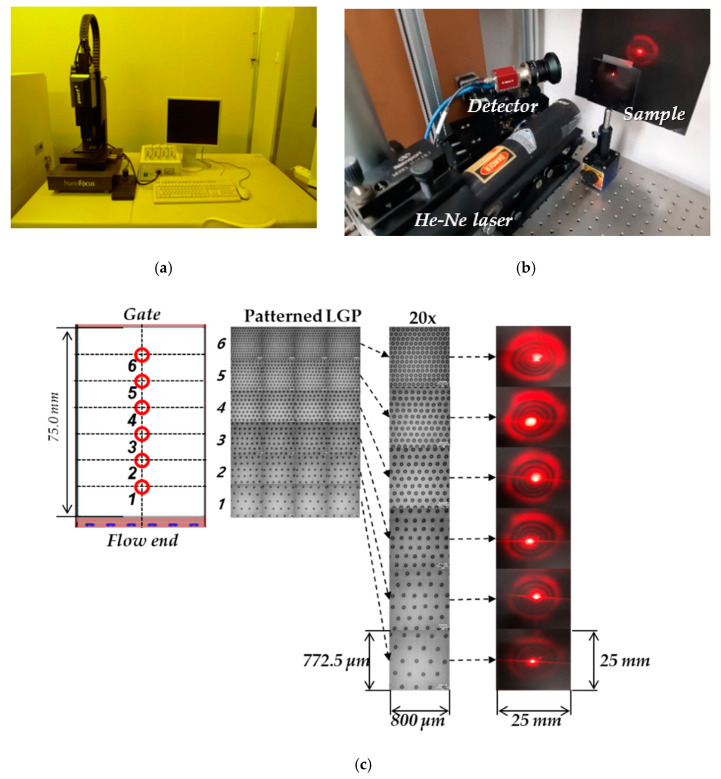
(**a**) A photograph of μ-surf (Nanofocus, Inc.). (**b**) Vision equipment for measuring scattering and diffraction patterns. (**c**) Micropattern on LGP surface and scattering image. (**d**) Conceptual set-up for industrial inspection system of injection-molded LGPs.

**Table 1 polymers-12-02610-t001:** Material properties of Iupilon HL4000 (Mitsubishi Co. Ltd.).

Properties		ISO Standard	Condition	Unit	Value
Physical	Specific gravity	ISO 1183		250	ton
	MFR	ISO 1133	300 °C, 1.2 kg	g/10 min	72
Mechanical	Tensile modulus	ISO527		MPa	2300
	Flexural modulus	ISO178		MPa	2500
Thermal	Heat deflection temp.	ISO75-2/A	1.8 MPa	°C	123
Optical	Total light transmittance	JIS-K7361		%	90.11
	Refractive index	MEP			1.59

**Table 2 polymers-12-02610-t002:** General specifications of injection molding machine (ROBOSHOT α-S250iA, FANUC).

Item	Value	Unit
Clamping force	250	ton
Screw diameter	32	mm
Max. injection speed	1200	mm/s
Max. injection pressure	3800	bar
Injection acceleration	5.02	G

**Table 3 polymers-12-02610-t003:** Process conditions and levels for molding trial.

	Level 1	Level 2	Level 3
Melt temperature (°C)	340	360	380
Injection speed (mm/s)	400	600	800
Mold temperature (°C)	70	80	90
Packing pressure (MPa)	30	40	50

**Table 4 polymers-12-02610-t004:** Taguchi L9 (3^4^) orthogonal array for design of experiments (DOE) of patterned and patternless LGP.

Case No.	Melt Temp.	Inj. Speed	Mold Temp.	Packing
1	1	1	1	1
2	1	2	2	2
3	1	3	3	3
4	2	1	2	3
5	2	2	3	1
6	2	3	1	2
7	3	1	3	2
8	3	2	1	3
9	3	3	2	1

**Table 5 polymers-12-02610-t005:** Performance specifications of Cary 5000 UV-Vis-NIR spectrometer.

Photometric System	Double Beam
Light source	Halogen lamp (visible), deuterium lamp (UV range)
Detectors	R928 PMT(UV-Vis)
Limiting resolution	<0.048 nm (UV-Vis)
Wavelength range	175–3300 nm
Wavelength accuracy	±0.08 nm at UV-Vis (190–900 nm)
Photometric accuracy	<0.00025 Abs (at 0.3 Abs UV-Vis)

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
