# Peer review of "An Experimental Study on Color Shift of Injection-Molded Mobile LGP Depending on Surface Micropattern"

_polymers, 2020, doi:10.3390/polym12112610_

Round 1

Reviewer 1 Report

  1. Line 12-13: “Conventional injection methods and injection processes have become more complex and harsher to mold these light guide plates.” It is necessary to describe in more detail what is the complexity of these processes for molding light guide plates.
  2. Line 15-16: „In the present study, injection molding experiments were conducted to understand the effect of injection molding process and surface  patterns on the color shift in injection-molded mobile LGP.” Here and in the relevant sections of the article, it is necessary to clarify what process parameters are in question. The same remark applies to line 22 - "all cases", line 53 -“various molding processes”.
  3. Line 23: What is low and medium density?
  4. Line18: CIE xy; Line 31: CRT and OLED; line 94 LGP; line 117 L9 must be deciphered.
  5. Line 32 Start your sentence with a capital letter.
  6. The percent sign (%) is on the left side of the equation (8)
  7. Subsections of the article would be better done as in the traditional format: experimental part, results and discussions.
  8. Table 4 is not clear what the numbers given in it mean
  9. In fig8, the inscriptions are not visible.
  10. It is very difficult to track changes using tables 6-8.
  11. The causes of yellowing and color shift are not discussed in detail, although this is a fundamental issue for this study.
  12. Figures 11b,d are missing

Author Response

Dear Reviewer,

I would like to give appreciation for your sincere reviews and detail comments.

We have made revisions based on your kind comments as follows.

  1. Comment: Line 12-13: “Conventional injection methods and injection processes have become more complex and harsher to mold these light guide plates.” It is necessary to describe in more detail what is the complexity of these processes for molding light guide plates.
  • Answer: Correction was made in abstract as follows.
  • Previous: Conventional injection methods and injection processes have become more complex and harsher to mold these light guide plates.
  • Corrected (page 1, line 12-14): Conventional injection methods and injection processes are becoming more complex like the ICM (Injection compression molding) and harsher with high speed injection at high mold and melting temperature.

  1. Comment: Line 15-16: “In the present study, injection molding experiments were conducted to understand the effect of injection molding process and surface patterns on the color shift in injection-molded mobile LGP.” Here and in the relevant sections of the article, it is necessary to clarify what process parameters are in question. The same remark applies to line 22 - "all cases", line 53 -“various molding processes”..
  • Answer: Correction was made in abstract as follows.
  • Previous: In the present study, injection molding experiments were conducted to understand the effect of injection molding process and surface patterns on the color shift in injection-molded mobile LGP.
  • Corrected (page 1, line 16-18): In the present study, injection molding experiment was conducted to understand the effect of surface patterns and major injection process conditions like mold and melt temperatures on the color shift in injection-molded mobile LGP.
  • Previous: all case --> Corrected: all 9 experimental cases (page 1, line 23)
  • Previous: various molding processes --> Corrected: molding processes related to melt temperature, injection speed, mold temperature, injection speed, packing pressure (page 2, line 55-56)

  1. Comment: Line 23: What is low and medium density?
  • Answer: "low and medium density" means the density of the pattern of the light guide plate (LGP). LGP is designed with a pattern density for uniform luminance from the LED's input to the end of the light. We added a sentence for understanding as below.
  • Previous: low and medium density --> Corrected: low and medium density of the pattern for the LGP (page 1, line 25)
  1. Comment: Line18: CIE xy; Line 31: CRT and OLED; line 94 LGP; line 117 L9 must be deciphered.
  •  Answer: Provided all abbreviations in their full names when they first appear.
  • Previous: CIE xy --> Corrected: CIE xy chromaticity diagram (page 1, line 19)
  • Previous: CRT and OLED --> Corrected: CRT (Cathode Ray Tube) to OLED (Organic Light Emitting Diode) (page 1, line 33-34)
  • Previous: LGP --> Corrected: It was already mentioned earlier. (page 1, line 11, 37)
  • Previous: L9 --> Taguchi L9 (34) (page 5, line 120)

  1. Comment: Line 32 Start your sentence with a capital letter.
  • Answer: Corrected to start with a capital letter.
  • Previous: but --> Corrected: But (page 1, line 35)

  1. Comment: The percent sign (%) is on the left side of the equation (8)
  • Answer: The percent sign (%) was moved on the right side of the equation (8)
  • Previous: %T --> Corrected: T (%) (page 3, equation (8))

  1. Comment: Subsections of the article would be better done as in the traditional format: experimental part, results, and discussions.
  • Answer: We moved the article and figures as follows according to your suggestion. Several Figure numbers have been changed due to the location movement of the article and figure related to previous Figure 3.
  • Previous: Figure 3 ---> Corrected: Figure 5 (page 6, line 125)
  • Previous: Figure 4 --> Corrected: Figure 3 (page 4, line 105)
  • Previous: Figure 5 --> Corrected: Figure 4 (page 4, line 106)

  1. Comment: Table 4 is not clear what the numbers given in it mean
  • Answer: The L9 orthogonal array is meant for understanding the effect of 4 independent factors each having 3 factor level values. This array assumes that there is no interaction between any two factors. Readers can refer to the level of each number in Table 3. (page 5, Table 4)

  1. Comment: In Fig 8., the inscriptions are not visible.
  • Answer: Inscriptions have been resized and repositioned to make them easier to see. (page 8, Figure 8.)

  1. Comment: It is very difficult to track changes using tables 6-8. (à table 5-8)
  • Answer: It was expressed to help the reader understand and show detailed data, but as the reviewer mentioned, the reader can understand it with graphs.
  • Previous: Table 5 ~8 --> Corrected: we decided to delete Table 5-8 as follow at the suggestion of all reviewers. (Deleted)

  1. Comment: The causes of yellowing and color shift are not discussed in detail, although this is a fundamental issue for this study.
  • Answer: Please look at line 223 to line 230 on page 11. This experimental study revealed that the cause of yellowing and color change in LGP is caused by scattering or other optical phenomena by the injection molding process and pattern. And Min et al. describes the color shift under different process conditions in 4.2. [13].

  1. Comment: Figures 11b,d are missing
  • Answer: Perhaps there are losses in Figures 11(b) and 11(d) during the upload process. You will be able to see them in newly uploaded file. (page 10)

Reviewer 2 Report

In this work, injection molding experiments were conducted to understand the effect of injection molding process and surface patterns on the color shift in injection-molded mobile LGP. Optical properties obtained by the direct and total transmittance and CIE xy of LGP for mobile injection-molded were analyzed. 

I therefore recommend this paper for publication, after the authors consider adding revisions based on the following suggestions:

  1. What are the optical properties of light guide plate (LGP)? They are only yellowing and color shift or something else? Whether the “optical characteristics” in the topic can change to “color shift”?
  2. I think the article is lack of summary conclusion and further analysis with the experimental results.
  3. What’s the standard for choosing the process conditions in Table 3? There is nothing introduction in Introduction about the effecting of process conditions on the molding parts, but in your results, the opinion of “total or direct transmittance was not affected by molding process variables” was presented. The question is: is it possible that the selection of process condition was wrong?
  4. In introduce of section 4.1. , only the case No.1, 4 and 7 were chosen compared with all 9 cases in section 4.2, then the conclusion of total or direct transmittance was not affected by molding process variables was drawn, which was not sufficiently.
  5. Is it necessary to list table 5,6,7,8?
  6. Fig.11 (b)? Fig.11 (d)?
  7. Line 57 the expression “the existence of surface micro pattern” is not consistent with that in line 16.
  8. Line 98 the definition of dimension direction “79.08mm in height, 51.72mm in length” is not consistent with that in Fig.4 (b).

    9. It is not a very good statement in line 205.

Author Response

Dear Reviewer,

I would like to give appreciation for your sincere reviews and detail comments.

We have made revisions based on your kind comments as follows.

Thank you.

Reviewer 3 Report

Basically the paper is ok, but it contains a whole bunch of small points to be corrected. Please have yourself a look through your paper to correct, since I may have not found them all.

For all your expermental data add the variation/standard deviation in the figures

Line 3: 'Depending'

line 20 - 26: these results are to much for the abstract. Here you can place something like in the last part of your conclusions.

line 31: you use CRT as an abbreviation for cathode ray tube. Please check your paper that every abbreviation should also be given in the full name at its first occurance.

line 32: delete the point

line 34: 'Figure 1'

line 38: 'Ju'

line 43: 'trend to thinner displays'; 'Yokoi'

line 45: 'Hong et. al.'

line 71: 'Here'

line 114: what do you want to say with this sentence?

line 154: 'total'

line 160:'value as in'

line 161: 'patterned'

line 167: 'shift, whether'

line 175: I think you mean 'direct transmittance' here since the total one is shown in Fig. 7.

Tables 5 to 8: you can omit them since they give no additional information to the final result/conclusion.

line 196 and 202: reference to Fig. 8 would fit more here since you are discussion the direct transmittance.

line 202: 'end (Point'

line 203: 'area (Point'

line 213 and in general: for the equipment used provide the type and manufacturer of it.

line 222: 'can be greater'

In Fig. 11 two pictures are missing (b and d). Please check the discussion of Fig. 11 regarding the clarity for the reader.

line 233: T1, etc is too detailed here. Give the general point which you want to express.

Author Response

(The authors gave the same response as above.)

Round 2

Reviewer 1 Report

The manuscript warrants publication in Polymers.

Reviewer 2 Report

Accepet in present form.